# A Generalized Method for the Synthesis of Carbon-Encapsulated Fe_3_O_4_ Composites and Its Application in Water Treatment

**DOI:** 10.3390/molecules27206812

**Published:** 2022-10-11

**Authors:** Shengxia Duan, Jian Liu, Yanling Pang, Feng Lin, Xiangyan Meng, Ke Tang, Jiaxing Li

**Affiliations:** 1Department of Chemistry and Engineering, Heze University, Heze 274500, China; 2CAS Key Laboratory of Photovoltaic and Energy Conservation Materials, Institute of Plasma Physics, Chinese Academy of Sciences, Hefei 230031, China; 3Department of Agriculture and Bioengineering, Heze University, Heze 274000, China

**Keywords:** α-Fe_2_O_3_, C@Fe_3_O_4_ composites, ferromagnetic, MB adsorption

## Abstract

In this paper, a simple and environmentally friendly method was developed for the preparation of highly stable C@Fe_3_O_4_ composites with controllable morphologies using sodium alginate as the carbon source and the easily obtained α-Fe_2_O_3_ as the precursors. The morphologies of the as-prepared C@Fe_3_O_4_ composites, inherited from their corresponding precursors of α-Fe_2_O_3_, survived from the annealing treatments, were characterized by the field-emission scanning electron microscopy (FESEM), transmission electron microscopy (TEM), X-ray diffraction (XRD) and inductively coupled plasma-atomic emission spectroscopy (ICP-AES). The C@Fe_3_O_4_ composites resisted to oxidation, acidification and aggregation, exhibiting porous structures and ferromagnetic properties at room temperature. Moreover, the adsorption performance of the C@Fe_3_O_4_ composites was evaluated by absorbing MB (methylene blue) in liquid environment. Experiments indicated that the C@Fe_3_O_4_ composites exhibited highly enhanced adsorption capacities and efficiencies as compared with their corresponding precursors of α-Fe_2_O_3_. This generalized method for the synthesis of C@Fe_3_O_4_ composites provides promising applications for the highly efficient removal of MB from industrial effluents.

## 1. Introduction

Among the various kinds of contaminants causing water pollution, dye pollution is becoming more serious in our environment and has become the focal point of research regarding the future of wastewater treatment, since organic dyes have been ubiquitously utilized in various fields, including plastics, leather, textiles, and cosmetics, etc. [1,2]. However, the unrestricted applications of these organic compounds result in serious environmental pollution because of their resistance to decomposition [3,4]. Particularly, as a representative organic dye, the removal of methylene blue (MB) from wastewater is an environmental issue and a lot of research studies have been carried out in this field. MB is a heterocyclic aromatic compound with extensive use in the textile industry and it is also present in the effluents of wastewater coming from other industries [5]. The acute exposure to MB may cause some harmful effects such as increased heart rate, shock, vomiting, jaundice, and tissue necrosis in humans [6,7]. Hence, it is essential to remove MB from industrial wastewater. Among various physical and chemical approaches, the adsorption technique is the most widely used method because of its simplicity, flexibility, cost-effectiveness and high efficiency for the dye removal [8,9,10]. Recently, magnetic adsorbent materials have gained special attention in water purification, based on their numerous advantages such as high separation efficiency, simple manipulation process, and mild operation conditions [11,12,13,14,15]. As one of the spinel ferrites, Co_0.5_Zn_0.5_Fe_2_O_4_ nanoparticles are significant magnetic nano-adsorbents because of their excellent surface properties and tunable size. Hence, Lv and his colleagues synthesized the magnetic Co_0.5_Zn_0.5_Fe_2_O_4_ nanoparticles via the facile nitrate-alcohol solution combustion process, which was further applied to remove methyl blue (MB) from an aqueous medium with its maximum adsorption capacity of 98.5 mg∙g^−1^ [11]. Moreover, because of the excellent magnetism, electrical property, and adsorption capacity, Liu and his co-workers also prepared the magnetic Ni_0.3_Co_0.4_Zn_0.3_Fe_2_O_4_ nanoparticles through a rapid-combustion process to adsorb MB from aqueous solution, and its maximum adsorption capacity was up to 101.1 mg∙g^−1^ [12]. In addition, to achieve the rapid recovery of the adsorbent by an external magnetic field, Zhou et al. have reported the adsorption of MB using synthesized lignin-based magnetic Fe_3_O_4_ nanoparticles and the maximum adsorption capacity of MB dye was 234.27 mg∙g^−1^ [13]. Similarly, to take advantage of the simple magnetic separation, Iram et al. also synthesized the Fe_3_O_4_ hollow nano-spheres via a one-pot simple template-free hydrothermal method to remove neutral red dye contaminants from the aqueous solution [14]. Additionally, Sharma and his co-workers have also applied the magnetic Fe_3_O_4_ nanoparticles to remove the new coccine from aqueous solutions to achieve the rapid recovery of the adsorbent by an external magnetic field [15].

Although these ferrite materials based on Fe_3_O_4_ nanoparticles have drawn considerable attention in wastewater treatment, they turned out not to be of practical application because of some drawbacks. It is well known that the Fe_3_O_4_ is highly susceptible to oxidative and acid conditions and can easily agglomerate into bigger clusters, which result in losing the specific properties of the magnetic nanoparticles and decreasing adsorption capacities [16]. Hence, covering Fe_3_O_4_ nanoparticles with a protective layer was beneficial to improving its stability. One of the most effective and feasible approaches is to cover the Fe_3_O_4_ nanoparticle’s surface with a carbon layer, which could protect it from being oxidized and reducing the magnetic interactions of magnetic nanoparticles.

To date, a series of carbon sources have been utilized to prepare carbon-encapsulated magnetic nanoparticles. For instance, Wang et al. synthesized the Fe_3_O_4_@C nanoparticles through an in situ solid-phase method to decolorize high concentrations of MB [17]. Kong et al. successfully synthesized Fe_3_O_4_@C nanoparticles using glucose as the carbon resource to fabricate an adsorbent for the removal of MB and cresol red (CR) [18]. Ai and his co-workers have prepared the graphene nanosheet (GNS)/magnetite (Fe_3_O_4_) composite by in situ converting FeCl_3_ to Fe_3_O_4_ and reducing graphene oxide (GO) into graphene simultaneously in an ethylene glycol solution for the removal of MB [19]. Banerjee and his colleagues also prepared a porous adsorbent of a carbon-Fe_3_O_4_ nanocomposite by using benzene dicarboxylic acid (BDC) as a carbon resource for the removal of MB using [20]. Although these carbon-coated ferromagnetic nanoparticles have been well developed, their practical applications are still limited for all kinds of reasons, such as high pollution, high-energy consumption, tedious synthetic steps, and low yields during the fabrication process. Thus, to meet the growing need of C@Fe_3_O_4_ composites, searching for suitable and “green” carbon sources is still significant to assure safe reactants and to avoid environmental pollution in the preparative process.

Currently, as far as we know, investigations on the preparation of C@Fe_3_O_4_ compound materials were rarely carried out using sodium alginate as the carbon source which is an inexpensive, non-toxic and renewable bio-resource belonging to the concept of “green chemistry”. Moreover, α-Fe_2_O_3_, which takes the role of the most stable ferrite oxides in an environmental condition, has been extensively applied in many fields [21], such as adsorption [22,23], catalysis [24,25], sensors [26], nanofluids [27], and lithium-ion batteries [28,29] because of its low cost, safety, and it is environmentally benign. Therefore, to protect Fe_3_O_4_ from being oxidized and applying suitable and “green” carbon sources, a simple and environmentally friendly method has been developed for the preparation of highly stable dispersions of C@Fe_3_O_4_ composites with controlled morphologies, including nanorods, nanodisks, and nanoparticles, using sodium alginate as the carbon source and the easily obtained α-Fe_2_O_3_ as the precursors. The α-Fe_2_O_3_ precursors were successfully prepared through a simple hydrothermal method, followed by an ionic polymerization to produce α-Fe_2_O_3_/carbon shell beads. The dried up α-Fe_2_O_3_/carbon shell beads obtained above were calcined at 400 °C for 2 h in a tube furnace and acid treatment, obtaining the high yields of C@Fe_3_O_4_ composites. These C@Fe_3_O_4_ composites exhibit the similar morphologies as their precursors as well as a ferromagnetic characteristic at room temperature with a higher coercivity (*Hc*) value. A high-enhanced adsorption capacity and efficiency for the removal of MB were also observed in comparison with the precursors of α-Fe_2_O_3_. In particular, the commercially available α-Fe_2_O_3_ materials were also applied to serve as the precursor mainly based on the following considerations. Firstly, it is used to verify whether this method can be generally applied to prepare similar C@Fe_3_O_4_ composites with different kinds of precursors. Secondly, it is used to compare the adsorption performance of C@Fe_3_O_4_ composites for MB removal by using different kinds of precursors and further to confirm the morphology effect on the adsorption performance.

## 2. Results and Discussion

### 2.1. Properties of the Synthesized Samples

XRD data were obtained to establish the composition and phase purity of the as-prepared α-Fe_2_O_3_ precursors, as shown in Figure 1a. The XRD pattern of α-FeOOH confirms that the initial yellow products after the hydrothermal synthesis are pure orthorhombic phase goethite iron oxide (α-FeOOH, JCPDS No. 81-0463). At the same time, the cell parameters of α-FeOOH, obtained through the least square fitting method, are *a* = 4.596 Å, *b* = 9.837 Å, and *c* = 3.197 Å, which further corroborate a similar structure as α-FeOOH (JCPDS No. 81-0463). The position and relative intensity of the reflection peaks of the α-Fe_2_O_3_ nanorod and nanodisk confirm the structure of the hematite, which agree well with the XRD pattern of α-Fe_2_O_3_ (JCPDS No. 86-2368) with structural parameters of *a* = *b* = 5.035 Å, *c* = 13.74 Å, indicating the high purity of the samples. The characteristic peaks can be identified at 2θ = 24.32°, 33.4°, 35.76°, 41.08°,49.56°, 54.2°, 57.84°, 62.52°, and 64.16°, which correspond to their indices of (0 1 −1 2), (1 0 −1 4), (1 1 −2 0), (1 1 −2 3), (0 2 −2 4), (1 1 −2 6), (1 2 −3 2), (2 1 −3 4), and (3 0 −3 0), respectively. Figure 1b,c shows the SEM images of iron-based samples. It can be clearly seen that all these samples have uniform and smooth surfaces. The diameter of the α-FeOOH nanorod is measured to be 120 nm, while 110 nm for the α-Fe_2_O_3_ nanorod, which can be attributed to the annealing treatment, making the structure of α-Fe_2_O_3_ nanorod much tighter. The disk-like structure of α-Fe_2_O_3_ is observed with a diameter of around 175 nm, as depicted in Figure 1d.

The size and morphology of the iron-based samples were further examined using transmission electron microscopy (TEM). Figure 2a also clearly displays a rod-like morphology with its porous structure of the α-FeOOH nanorod, of which the lattice fringes are measured by HRTEM (Figure 2d) with d-spacings of 0.261 and 0.262 nm, corresponding to the spacings of the (0 −2 1) and (0 2 1) planes of α-FeOOH, respectively. Similarly, the porous structure of the α-Fe_2_O_3_ nanorod is clearly shown in Figure 2b, as well as the Brunauer-Emmett-Teller (BET) measurements in Appendix A. The corresponding disk-like structure of the α-Fe_2_O_3_ nanodisk is also depicted in Figure 2c. The representative HRTEM images taken from the α-Fe_2_O_3_ nanorod and nanodisk sample are shown in Figure 2e,f, which distinctly indicate that the preparative α-Fe_2_O_3_ samples are compounded from many fine, crystallized particles with different crystal planes and orientations. The lattice fringes from both α-Fe_2_O_3_ samples can be clearly observed with d-spacings of 0.267 and 0.259 nm, corresponding to the spacings of the (1 0 −1 4) and (1 1 −2 0) planes, respectively. Moreover, both angles are observed to be 60°, which is consistent with the theoretical calculation between the (1 0 −1 4) and (1 1 −2 0) planes.

Figure 3a shows the XRD patterns of these C@Fe_3_O_4_ composites, i.e., nanorod, nanodisk, and nanoparticles. All the diffraction peaks can be indexed to the face-centered cubic structure of magnetite according to JCPDS No. 89-0691 with structural parameters of *a* = *b* = *c* = 8.394 Å. The diffraction peaks were indexed at 2θ values of 18.44° (1 1 1), 30.20° (2 2 0), 35.60° (3 1 1), 43.24° (4 0 0), 53.52° (4 2 2), 57.20° (5 1 1), and 62.8° (4 4 0). All C@Fe_3_O_4_ composites are highly crystallized, confirmed by the observation of the strong and sharp peaks in Figure 3a. Moreover, there are no other diffraction peaks that have originated from other phases, suggesting that the as-synthesized samples have higher purity. However, due to the similar patterns between γ-Fe_2_O_3_ and Fe_3_O_4_, the XRD analyses cannot provide enough evidence to confirm the transformation [30]. Thus, the Raman measurements are provided for further structural confirmation, which is also a frequently used and non-ruinous method to characterize carbonaceous material, especially applied to certify the degree of graphitization [31]. The corresponding Raman spectra are shown in Figure 3b–d. The weak peaks at 666, 667, and 655 cm^−1^ of three C@Fe_3_O_4_ composites are attributed to the A_1g_ mode of magnetite, according to Phase et al. [32,33], suggesting that the main component of C@Fe_3_O_4_ composites is Fe_3_O_4_, although no visible peak is recorded at 540 cm^−1^. The peaks at 1372, 1385, and 1400 cm^−1^ are associated with the vibrational modes of carbon atoms with dangling bonds for the in-plane-terminated and disordered graphite, labeled as the D band. The corresponding G band peaks are found at 1594, 1594, and 1595 cm^−1^, corresponding to the E_2g_ mode for the vibration of all sp^2^-bonded carbon atoms in a two-dimensional (2D) structure [31]. The degree of graphitization can be ascertained from the *I_D_*/*I_G_* value (relative intensity ratio of the D-G band) [16]. Furthermore, the *I_D_*/*I_G_* values of the three samples are all approximately 0.8, suggesting that these carbon materials have a high degree of disordered graphitization, which is coincident with the XRD analysis results.

Similarly, the structures of the C@Fe_3_O_4_ composites were also characterized by SEM and TEM observations. As shown in Figure 4a–c, it can be clearly observed that the morphologies of the precursors were not damaged during the process of calcination. However, the structures of the C@Fe_3_O_4_ composites became much tighter after annealing treatment in comparison to that of the precursors, which can be attributed to the presence of the carbon layer on the surface of the C@Fe_3_O_4_ composites. Meanwhile, the surfaces of the C@Fe_3_O_4_ composites are getting rougher in comparison with pure α-Fe_2_O_3_. Magnified TEM images further demonstrate that the Fe_3_O_4_ particles are surrounded by a thick carbon layer (Figure 4d–f). Thus, the advent of the broad XRD peaks in Figure 3a is ascribed to the presence of the nanosized Fe_3_O_4_ modified by a carbon layer. A similar phenomenon was also found in previous research [34].

The HRTEM images of C@Fe_3_O_4_ composites were also presented in Figure 5 to compare with that of their corresponding precursors. For the C@Fe_3_O_4_ nanorod and nanodisk, the lattice fringes with d-spacings of 0.297 and 0.266 nm are clearly observed, which correspond to the spacings of the (2 2 0) and (0 0 3) planes of Fe_3_O_4_, respectively (Figure 5a,c). Both lattice fringe angles are observed to be 90°, which is consistent with the theoretical calculation of the angle between the (2 2 0) and (0 0 3) planes. For C@Fe_3_O_4_ nanoparticles (Figure 5c), the typical lattice spacings are observed as 0.297 and 0.251 nm, corresponding to the lattice distance of (2 2 0) and (1 1 3) planes, respectively. Moreover, the lattice spacing of the carbon layer (approximately 0.207 nm) is also presented in Figure 5b,d,f, which can be attributed to the (1 1 1) plane of the carbon layer. The existence of carbon was further corroborated by the energy dispersive X-ray spectroscopy (EDS) analysis (Appendix A).

BET gas sorptometry measurements were conducted to examine the porous nature of these C@Fe_3_O_4_ composites. Figure 6 shows the N_2_ adsorption/desorption isotherm and the pore-size distribution of the C@Fe_3_O_4_ composites (a) nanorod, (b) nanodisk, and (c) nanoparticle, respectively. The isotherms of the three C@Fe_3_O_4_ composites clearly indicate their characteristic mesoporous structure, identified as type IV. The specific BET surface areas of nanorod, nanodisk, and nanoparticles are 31.407 m^2^·g^−1^, 26.682 m^2^·g^−1^ and 27.003 m^2^·g^−1^, respectively, which are a little larger than their corresponding precursors of α-Fe_2_O_3_ (14.584 m^2^·g^−1^, 9.229 m^2^·g^−1^ and 13.517 m^2^·g^−1^, see Appendix A). The pore-size distribution is obtained from BJH analysis, which shows numerous pores with less than 10 nm in these C@Fe_3_O_4_ composites, thus, strongly supporting the mesoporous structures of these C@Fe_3_O_4_ composites. The pores in both C@Fe_3_O_4_ nanorod and C@Fe_3_O_4_ nanodisk originated from the porous nature of the materials themselves, while the pores in C@Fe_3_O_4_ nanoparticles originated from aggregates among a lot of small nanoparticles obtained from the observation of TEM images of the three C@Fe_3_O_4_ composites. Moreover, as observed from the BJH pore-size distribution curve (Figure 6 and Appendix A), the sharp-pointed hole size distribution is approximately 4 nm for C@Fe_3_O_4_ composites, while the pore sizes are approximately 20–30 nm in the precursors, which can be explained by the presence of the carbon layer generating smaller pore diameters and relatively larger surface areas [34], thus, enhancing the adsorption abilities of C@Fe_3_O_4_ composites for the removal of MB in solution, eventually. Above all, the TEM and BET characterizations clearly demonstrate that a hexagonal close-packed oxide ion array of α-Fe_2_O_3_ was successfully transitioned to the cubic close-packed array of Fe_3_O_4_ after the calcination process. Additionally, the porous structure of the sample was not destroyed even with the significant structural changes.

The magnetic properties of C@Fe_3_O_4_ composites were studied under ambient temperature (300 K) with an external magnetic field of −20,000 Oe to 20,000 Oe, as illustrated in Figure 7. The hysteresis loops of magnetization of three C@Fe_3_O_4_ composites show ferromagnetic behaviors. The saturation magnetization (*Ms*), remanent magnetization (*Mr*), and coercivity (*Hc*) values were shown in Table 1. It could be clearly seen that the saturation magnetizations (*Ms*) of C@Fe_3_O_4_ composites were lower in comparison to that of raw Fe_3_O_4_ (92 emu·g^−1^) [35]. However, their *Ms* values were higher than other formerly researched carbon-encapsulated Fe_3_O_4_ composites, for instance, Fe_3_O_4_–C microrods (0.91 emu·g^−1^), Fe_3_O_4_@C composites (13.0 emu·g^−1^) [36], and Fe_3_O_4_–C nanowires (5.11 emu·g^−1^) [37]. The decrease could be ascribed to the widespread presence of carbon layer on the Fe_3_O_4_ nanoparticles’ surfaces, leading to their relatively ordered interpenetration in the carbon layer, which would hinder the optional movement of Fe_3_O_4_ nanoparticles and weaken the magnetic interactions between these magnetic nanoparticles. In addition, the *Hc* values were higher in comparison to that of raw Fe_3_O_4_ nanoparticles. Nevertheless, these *Hc* values were lower that of the formerly researched Fe_3_O_4_-C coaxial nanofibers (324.5 Oe) [38]. This phenomenon can be explained based on the higher shape anisotropy and decreased particle size of the C@Fe_3_O_4_ composites. The higher shape anisotropy of the C@Fe_3_O_4_ composites will induce them to magnetize in the direction of the easy axis of magnetization. Since C@Fe_3_O_4_ composites were relatively randomly oriented in comparison to that of the raw Fe_3_O_4_ nanoparticles, the projections of the magnetization vector in the direction of the magnetic field will be lower in comparison to that of these raw Fe_3_O_4_ nanoparticles accompanied by higher shape anisotropy. Furthermore, the C@Fe_3_O_4_ composites investigated in this study have lower shape magnetic anisotropy in comparison to that of the Fe_3_O_4_–C coaxial nanofibers possessing single crystalline Fe_3_O_4_ cores, which, ultimately, brought about the decrease in *Hc* values. Hence, it can be concluded that the magnetic properties of the C@Fe_3_O_4_ composites were closely related to the features of the materials, such as particle size, morphology, and the characteristics of micro-pores [39].

### 2.2. Adsorption Kinetics of the C@Fe_3_O_4_ Composites

In this study, three C@Fe_3_O_4_ composites and their precursors were used to absorb MB from the aqueous solution. Figure 8 presents the time-dependent adsorption process of MB onto three C@Fe_3_O_4_ composites and their precursors of α-Fe_2_O_3_. The adsorption capacities of three C@Fe_3_O_4_ composites were observed to be 35.65 mg·g^−1^, 35.12 mg·g^−1^, and 35.35 mg·g^−1^ for nanorod, nanodisk, and nanoparticle, respectively. However, the adsorption capacities of their corresponding precursors for MB removal were only 0.368 mg·g^−1^, 0.342 mg·g^−1^, and 0.355 mg·g^−1^, respectively. Converting to the adsorption efficiencies, the values for the three C@Fe_3_O_4_ composites reached up to 90% in 360 min, while only approximately 0.9% for the α-Fe_2_O_3_ precursors. This phenomenon can be ascribed to the increased active binding sites on the surface of the C@Fe_3_O_4_ composites in comparison with their α-Fe_2_O_3_ precursors. Thus, the relatively fast adsorption rate, large adsorption capacity, easy magnetic separation, and environmental benign synthetic route of the C@Fe_3_O_4_ composites make them ideal adsorption and enrichment materials for MB removal.

To investigate the adsorption kinetic process and mechanism, four kinetic models were applied to fit the kinetic adsorption data. Firstly, the linear regressions of the pseudo-first-order kinetic and pseudo-second-order kinetic models were used to fit the adsorption kinetics data for MB adsorption onto the obtained samples [40,41]. The above two kinetic models were illustrated as Equations (1) and (2) for pseudo-first-order kinetic and pseudo-second-order kinetic models, respectively:(1)$$\log(q_{e}-q_{t})=\log q_{e}-\frac{k_{1}}{2.303}t$$
(2)$$\frac{t}{q_{t}}=\frac{1}{2k_{2}q_{e}^{2}}+\frac{1}{q_{e}}t$$
where *q_e_* and *q_t_* (mg⋅g^−1^) are the amounts of MB adsorbed at equilibrium and at time t (min), respectively. *k*_1_ (min^−1^) is the pseudo-first-order kinetic rate constant, which represents the adsorption rate when the adsorption process conforms to the pseudo-first-order kinetics equation. Additionally, *k*_2_ (g⋅mg^−1^⋅min^−1^) is the pseudo-second-order kinetic rate constant. Additionally, the fitting results were shown in Figure 9a,b, in company with Table 2. The value of the correlation coefficient (*R*^2^) obtained from pseudo-first-order kinetic model are relatively poor (0.9290, 0.9572, and 0.8641 for C@Fe_3_O_4_ nanorod, nanodisk, and nanoparticle composites, respectively), while the *R*^2^ values obtained from the pseudo-second-order kinetic model for the C@Fe_3_O_4_ nanorod, nanodisk, and nanoparticle composites, are 0.9996, 0.9985, and 0.9997. Moreover, the *q_e,cal_* of the pseudo-second-order model (36.36 mg·g^−1^, 34.0136 mg·g^−1^, and 35.1494 mg·g^−1^ of C@Fe_3_O_4_ nanorod, nanodisk, and nanoparticle composites, respectively) is very close to the experimental *q_e_* values. These results clearly indicate that the MB adsorption onto the C@Fe_3_O_4_ composites can be well described by the pseudo-second-order kinetic model and that chemical adsorption might be involved in the process. Moreover, in this study, the *k*_1_ values followed the order of 0.1531 (min^−1^) > 0.0477 (min^−1^) > 0.0182 (min^−1^) for C@Fe_3_O_4_ composites nanorod, nanoparticle, and nanodisk, respectively, which was consistent with the comparisons of adsorption performance for the above C@Fe_3_O_4_ composites. Although the MB adsorption onto the C@Fe_3_O_4_ composites could not be well described by the pseudo-first-order kinetics model due to the poor *R*^2^ values, it can still be inferred that the larger BET surface area, suitable morphology structures, and pore-size distributions would be beneficial for increasing the adsorption rate. Additionally, the *k*_2_ values followed the order of 0.0205 g·mg·min^−1^) > 0.0120 (g·mg·min^−1^) > 0.00085 (g·mg·min^−1^) for C@Fe_3_O_4_ nanorod, nanoparticle, and nanodisk composites, respectively, which was also consistent with the comparisons of adsorption performance for the above C@Fe_3_O_4_ composites, thus, further confirming the significant role of the BET surface area, morphology structures, and pore-size distributions in increasing the adsorption rate.

Secondly, the intra-particle diffusion model was utilized to obtain a more particular knowledge of adsorption data, and the equation was expressed as Equation (3) [42]:(3)$$q_{t}=k_{i}t^{1/2}+C$$
where *k_i_* (mg∙g^−1^∙min^−1^) is the rate constant of the intra-particle diffusion model and *C* (mg∙g^−1^) is a constant, which is proportional to the extent of boundary layer thickness. As shown in Figure 9c and Appendix A, the adsorption of MB onto the C@Fe_3_O_4_ composites can be divided into three typical processes, including external, interfacial, and eventual equilibrium processes. The first stage with the sharp increase in adsorption amount was mainly controlled by the outside surface adsorption. Additionally, in this stage, a great quantity of MB molecules was immobilized onto the adsorption site of the adsorbents. Subsequently, the MB molecules diffused into the pores of the adsorbents, which were adsorbed by the interior surface of pores, in which the intra-particle diffusion became the primary rate-determining step. The third step demonstrates the final equilibrium state. Moreover, none of the three fitting lines crossed the origin, suggesting that intra-particle diffusion is not the only rate-controlling step.

Furthermore, another kinetic fitting equation of the Boyd kinetic model was also utilized to confirm the practical rate-determining step regarding the MB immobilization onto the C@Fe_3_O_4_ composites. Additionally, the equation was illustrated as Equations (4)–(6) [43]:(4)$$F=1-\frac{6}{\pi^{2}}\exp(-Bt)$$
(5)$$F=\frac{q_{t}}{q_{e}}$$
(6)Bt=−0.4977−ln(1−F)
where *F* is donated as the fraction of solute, while parameter *B* is the mathematic function of *F*. As illustrated in Figure 9d and Appendix A, none of the three curves of the Boyd model plot crossed the origin, suggesting that liquid film diffusion was the rate-limiting step for MB immobilization onto the three C@Fe_3_O_4_ composites. Additionally, the fitting results achieved a good agreement with that of the intra-particle diffusion model. Furthermore, the kinetic fitting parameters of the immobilization process on the three α-Fe_2_O_3_ precursors were also evaluated by using these four kinetic models, as mentioned above (Appendix A).

### 2.3. Adsorption Kinetics of the C@Fe_3_O_4_ Composites

To investigate the maximum adsorption capacity of C@Fe_3_O_4_ composites for MB removal, the adsorption experiments on different initial concentrations of MB were carried out at room temperature and the results of the experiments were shown in Figure 10. It could be seen that the adsorption capacities of C@Fe_3_O_4_ composites were increased with the increase in the MB solution concentration. Particularly, the adsorption capacities of C@Fe_3_O_4_ composites increased rapidly in the situation of low initial concentrations, indicating that the C@Fe_3_O_4_ composites had a large number of active sites. Moreover, the adsorption capacities of the C@Fe_3_O_4_ composites became flat as the initial MB concentration went up, suggesting that the limited active sites were fully occupied by an excess of MB dye. Moreover, two isotherm models of Langmuir and Freundlich were applied to understand the type of interaction of adsorbents (C@Fe_3_O_4_ composites) with adsorbate (MB). The Langmuir isotherm was often applied to express the monolayer adsorption on homogeneous and energetically uniform surfaces, while the Freundlich isotherm model was extensively applied to determine the multilayer adsorption course [44,45]. Additionally, the two models were expressed as Equations (7) and (8) for the Langmuir and Freundlich isotherm model, respectively:(7)$$q_{e}=\frac{Q_{max}bC_{e}}{1+bC_{e}}$$
(8)$$q_{e}=K_{f}C_{e}^{1/n}$$
where *q_e_* (mg·g^−1^) represents the adsorption capacity of MB onto the C@Fe_3_O_4_ composites under the balance situation, *C_e_* (mg·L^−1^) represents the MB concentration under the balance situation, *Q_max_* (mg·g^−1^) represents the maximum adsorption capacity MB onto the C@Fe_3_O_4_ composites, *b* (L·mg^−1^) is related to the adsorption energy constant, *K_f_* represents the indicator of adsorption quantity, and 1/*n* represents the adsorption intensity.

The fitting results of the two above models were illustrated in Figure 10 and Table 3, which indicated that the Langmuir model yields a better fit for the experimental equilibrium adsorption data than the Freundlich model based on the comparison of the correlation coefficients (*R*^2^) of the two equations. This fact indicates that the binding energy is uniform on the whole surface of these C@Fe_3_O_4_ composites, suggesting a mono-layer coverage of MB on the surface of the C@Fe_3_O_4_ composites. Furthermore, another data parameter of *R_L_* (*R_L_* = 1/(1 + *bC*_0_)) was also applied to understand the adsorption process [46]. It could be also seen that all the *R_L_* values obtained from the above function of the MB adsorption onto the three C@Fe_3_O_4_ composites are all within the range from 0 to 1, suggesting a favorable immobilization procedure. Moreover, the Freundlich constant *n* is greater than 1, which reveals the favorable adsorption of MB onto the C@Fe_3_O_4_ composites. This is consistent with the conclusion of the Langmuir model. The maximum adsorption capacities (*Q_max_*) calculated from the above Langmuir model shown in Appendix A are 34.54 mg·g^−1^, 32.94 mg·g^−1^, and 33.64 mg·g^−1^ for the C@Fe_3_O_4_ nanorod, nanaodisk and nanoparticle composites, respectively. Generally speaking, suitable morphology structures, suitable pore-size distribution, and large specific surface area will be beneficial to promoting their adsorption properties in the process of dye adsorption [47,48]. The C@Fe_3_O_4_ nanorod owned the largest adsorption capacity, while the C@Fe_3_O_4_ nanodisk owned the lowest adsorption capacity, which can be explained as follows: Comparing with the morphology structures of the three C@Fe_3_O_4_ composites, as shown in Figure 4, the C@Fe_3_O_4_ nanodisks are stuck much tighter and a thicker carbon layer is wrapped outside of the cubic Fe_3_O_4_, making it difficult for the MB molecules to combine with the active sites, thus, decreasing the adsorption capacity of the C@Fe_3_O_4_ nanodisk. The C@Fe_3_O_4_ nanorod sample owns a relatively thin carbon layer wrapped outside of the cubic Fe_3_O_4_ (Figure 4d), making it easier for the MB molecules to combine with the active sites. Moreover, the C@Fe_3_O_4_ nanorod exhibited a larger surface area than the C@Fe_3_O_4_ nanodisk, and nanoparticle, which is further beneficial to its adsorption process toward MB adsorption. The difference in adsorption capacity is the result of the morphology structures and BET surface area, but has little connection with pore size, since the three C@Fe_3_O_4_ nanorod, nanodisk, and nanoparticle composites have similar pore sizes.

## 3. Experimental

### 3.1. Chemicals

All the reagents were of analytical grade and used without further purification including FeCl_3_·6H_2_O (ferric chloride hexahydrate), C_6_H_8_O_7_ (citric acid), NaOH (sodium hydroxide), Na_2_B_4_O_7_·10H_2_O (sodium tetraborate), CaCl_2_ (calcium chloride anhydrous), and sodium alginate. Deionized water was applied in this study.

### 3.2. Synthesis of C@Fe_3_O_4_ Nanorods

C@Fe_3_O_4_ nanorods were fabricated through a three-step process. (1) α-Fe_2_O_3_ nanorods were first obtained by a simple method. Specifically, 1.0 mmol of FeCl_3_ and 2 mmol of citric acid were added into 40 mL of water under vigorous stirring. A total of 25 mmol of NaOH(s) was added into the above solution while stirring for approximately 10 min. After that, the whole mixture was transferred into a Teflon-lined stainless-steel autoclave (LC-KH-50, Lichen Instrument Technology Co., Ltd., Shaoxing, China) for hydrothermal treatment at 140 °C for 12 h. As the autoclave cooled to room temperature naturally, the precipitates (α-FeOOH) were separated by centrifugation (KL02A, Hunan Kaida Scientific Instrument Co., Ltd., Changsha, China), washed with distilled water and absolute ethanol, and dried in vacuum at 60 °C for 12 h. Finally, to obtain the α-Fe_2_O_3_ nanorods, the as-prepared α-FeOOH was subjected to calcination in a muffle furnace (SX4-2-12, Sanfu Hezhong Technology Development Co., Ltd., Beijing, China) at 400 °C for 2 h with a temperature increasing rate of 1 °C·min^−1^. (2) The α-Fe_2_O_3_ nanorods/alginate composite beads were obtained through an ionic polymerization method. Particularly, sodium alginate powder with 1.5 g was stirred continuously in distilled water with 100 mL at 353.15 K for 12 h, gaining a viscous, transparent, and homogeneous sample solution. Subsequently, α-Fe_2_O_3_ nanorods with 1.5 g was mixed into the above-mentioned solution and continually stirred for another 30 min. The above mixed solution containing sodium alginate and α-Fe_2_O_3_ nanorods was added dropwise to the 0.18 mol·L^−1^ CaCl_2_ solution with a syringe to obtain magnetic beads, which were then washed a few times by using distilled water and collected by drying in vacuum at 60 °C for 12 h, ultimately. A schematic illustration is shown in Figure 1 for the fabrication of α-Fe_2_O_3_/alginate beads. (3) The as-prepared α-Fe_2_O_3_/alginate beads were further heated at 400 °C for 2 h in a tube furnace with a heating rate of 1 °C·min^−1^ under Ar atmosphere. The sample was treated with acid solutions, washed a few times using distilled water and collected by drying in vacuum at 60 °C for 12 h, ultimately. After the above treatment, the C@Fe_3_O_4_ nanorods were obtained. The C@Fe_3_O_4_ nanodisk and C@Fe_3_O_4_ nanoparticle were also prepared using α-Fe_2_O_3_ nanodisk and commercially available α-Fe_2_O_3_ as the precursors through a similar procedure as mentioned above. The detailed synthetic procedures are available in the Appendix A.

### 3.3. Characterization

X-ray diffraction (XRD) patterns were obtained using a Philips X’Pert Pro Super diffractometer (Amsterdam, The Netherlands) with Cu Kα radiation (λ = 1.54178 Å). The Raman spectra were recorded at an ambient temperature on a SPEX 1403 spectrometer (Metuchen, NJ, USA) with an argon-ion laser at an excitation wavelength of 514.5 nm. The scanning electron microscope (SEM) was carried out using a Hitachi S-4800 (Tokyo, Japan), which was equipped with an X-ray energy dispersive spectrometer of GENESIS4000. The high-resolution transmission electron microscopy (HRTEM) was performed on JEOL 2010 (Tokyo, Japan) with an accelerating voltage of 20 kV. N_2_ adsorption was measured using the Brunauer-Emmett-Teller method (ASAP-2020, USA) at 77 K. A superconducting quantum interference device (SQUID, San Diego, USA) was applied to investigate the magnetic properties of the samples.

### 3.4. Adsorption Experiment

Groups of experiments were designed to investigate the kinetics and isotherms of MB adsorption onto the magnetic materials. The stock solution of MB with 0.01 mol·L^−1^ was firstly prepared and then paired to certain concentrations in the following experiments.

The isothermal adsorption experiments were carried out under different initial MB concentrations (8.3 × 10^−5^, 8.7 × 10^−5^, 8.9 × 10^−5^, 9.0 × 10^−5^, 9.1 × 10^−5^, 9.2 × 10^−5^, 9.3 × 10^−5^ and 9.5 × 10^−5^ mol∙L^−1^) with a constant amount of C@Fe_3_O_4_ composites of 0.8 g∙L^−1^ at 298.15 K. The adsorptive capacities of C@Fe_3_O_4_ composites for MB were calculated using the following formula:$$q_{e}=\frac{(C_{0}-C_{e})V}{m}$$
where *C*_0_ (mg·L^−1^) and *C_e_* (mg·L^−1^) are the initial and equilibrium concentration of MB in the solution, *V* (L) is the total volume of the suspension, and *m* (g) is the mass of adsorbent. To determine the adsorption efficiency, the kinetic experiments were performed by using different flasks containing 0.8 g·L^−1^ of C@Fe_3_O_4_ composites. At the desired time intervals, the magnetic C@Fe_3_O_4_ composites were separated by using a hand-held permanent magnet, and the supernatant was collected for concentration measurements. The absorbance of the MB dye was measured using a UV–visible spectrophotometer (Shimadzu UV-1800 spectrophotometer, Kyoto, Japan). The maximum wavelength of absorbance for the MB dye is λ_max_ = 664 nm. All the adsorption experiments were carried out under solution pH values of 6.0 ± 0.1 and a temperature of 298.15 K. Moreover, the adsorption isotherm experiments were carried out for 48 h on an oscillator to achieve an adsorption equilibrium. Furthermore, all adsorption experiments were carried out in triplicate in order to ensure the accuracy of the experimental data.

## 4. Conclusions

Herein, a series of C@Fe_3_O_4_ composites with three morphologies of nanorod, nanodisk, and nanoparticles were successfully fabricated via a three-step process, using the easily obtained α-Fe_2_O_3_ as the precursor. The α-Fe_2_O_3_ precursors were first fabricated, followed by the generation of α-Fe_2_O_3_/carbon shell beads using sodium alginate as a carbon source. The magnetic hysteresis loop measurement shows that the C@Fe_3_O_4_ composites display ferromagnetic properties at room temperature. The saturation magnetization values of the C@Fe_3_O_4_ composites are lower, while the coercivity (*Hc*) values of the C@Fe_3_O_4_ composites are higher in comparison to that of the raw Fe_3_O_4_ nanoparticles, which could be ascribed to the widespread presence of carbon layer, the shape anisotropy, and the decreased particle size of C@Fe_3_O_4_ composites. These C@Fe_3_O_4_ composites exhibit highly enhanced adsorption capacities and efficiencies for the removal of MB in comparison with their corresponding precursors. The removal efficiencies of MB for C@Fe_3_O_4_ composites are over 93.83% in 360 min, while that of α-Fe_2_O_3_ precursors are approximately 0.9%. Thus, this generalized method for the synthesis of C@Fe_3_O_4_ composites provides promising applications for the highly efficient removal of MB from industrial effluents.

## Data Availability

The data presented in this research are available on request from the corresponding author.

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
