# Peer review of "A Generalized Method for the Synthesis of Carbon-Encapsulated Fe3O4 Composites and Its Application in Water Treatment"

_molecules, 2022, doi:10.3390/molecules27206812_

Round 1

Reviewer 1 Report

In my opinion the paper with title:

“A generalized method for the synthesis of carbon-encapsulated, Fe3O4 composites and its application in water treatment”

should be published in the Molecules journal if the authors will make the recommended changes and corrections:

1.  Please arrange the chapters of the article in accordance with the Instructions for Authors.

2.  At Chapter 2.2. Synthesis of C@Fe3O4 nanorods, please specify the type and producer of autoclave for hydrothermal treatment, of centrifugation device and of muffle furnace.

3.  At Chapter 2.4. Adsorption experiment, please introduce the values ​​of initial concentrations of MB used in adsorption experiments, and other conditions of work: pH, temperature, contact time, stirring, etc.

4.  Please specify the values ​​of same parameters for kinetic studies, also.

5.  In the Figure 2 the images d, e and f are not clear enough. Please present a better resolution of these images and for the images in Figure 3 and 5, also.

6.  Please specify what represent the following parameters: the saturation magnetization (Ms), remanent magnetization (Mr), and coercivity (Hc), values ​​that were shown in Table 1.

7.  In Figure 9 a, c and d, both points and lines appear. In the legend it is not mentioned what the lines represent.

8.  Figure 10 shows three types of data, experimental (dotted), calculated according to the Langmuir model (continuous lines) and calculated according to the Freundlich model (broken lines). Please specify this in the figure(legend) and at the name of the figure.  In the legend, you must specify that those are the experimental data, and in the name of the figure, insert a few words about the experimental points.

Author Response

A generalized method for the synthesis of carbon-encapsulated, Fe3O4 composites and its application in water treatment” should be published in the Molecules journal if the authors will make the recommended changes and corrections:

  1. Please arrange the chapters of the article in accordance with the Instructions for Authors.

Reply: We appreciate your great comment about this issue. Based on your comment, we have arranged the chapters of the article. Thanks again.

  1. At Chapter 2.2. Synthesis of C@Fe3O4 nanorods, please specify the type and producer of autoclave for hydrothermal treatment, of centrifugation device and of muffle furnace.

Reply: Great thanks for your suggestions about this issue. The Teflon-lined stainless steel autoclaves were LC-KH-50, provided by Lichen Instrument Technology Co., Ltd. The centrifuge of KL02A was provided by Hunan Kaida Scientific Instrument Co., Ltd. The muffle furnace of FAITHFUL, SX4-2-12 was provided by Sanfu Hezhong (Beijing) Technology Development Co., Ltd. (Please see Line 126-127, 129 and 132 Page 3 marked with BLUE words in this manuscript). Thanks again.

  1. At Chapter 2.4. Adsorption experiment, please introduce the values of initial concentrations of MB used in adsorption experiments, and other conditions of work: pH, temperature, contact time, stirring, etc.

Reply: Great thanks for your comment about this issue. The initial concentrations of MB used in adsorption experiments were 8.3×10-5, 8.7×10-5, 8.9×10-5, 9.0×10-5, 9.1×10-5, 9.2×10-5, 9.3×10-5 and 9.5×10-5 mol∙L-1. All the adsorption experiments were carried out under solution pH values of 6.0±0.1 and temperature of 298.15 K. Moreover, the adsorption isotherm experiments were carried out for 48 hours on oscillator to achieve adsorption equilibrium. The corresponding descriptions were added into this manuscript and marked with BLUE words. (Please see Line 170-171, 183-184 Page 4 and Line 185-187, Page 5). Thanks again.

  1. Please specify the values of same parameters for kinetic studies, also.

Reply: Great thanks for your suggestions about this issue. k1 (min-1) is the pseudo-first-order kinetic rate constant, which represents the adsorption rate when the adsorption process conforms to the pseudo first order kinetics equation. In this study, the k1 values followed the order of 0.1531 (min-1) > 0.0477 (min-1) > 0.0182 (min-1) for C@Fe3O4 composites nanorod, nanoparticle and nanodisk, respectively, which was consistent with the comparisons of adsorption performance for the above C@Fe3O4 composites. Although the MB adsorption onto C@Fe3O4 composites could not be well described by pseudo first order kinetics model due to the poor R2 values, it can still be inferred that larger BET surface area, suitable morphology structures and pore size distributions would be beneficial increasing the adsorption rate. Moreover, k2 (g·mg·min-1) is the pseudo-second-order kinetic rate constant, which represents the adsorption rate when the adsorption process conforms to the pseudo second order kinetics equation. In this study, the k2 values followed the order of 0.0.0205 (g·mg·min-1) > 0.0120 (g·mg·min-1) > 0.00085 (g·mg·min-1) for C@Fe3O4 composites nanorod, nanoparticle and nanodisk respectively, which was also consistent with the comparisons of adsorption performance for the above C@Fe3O4 composites, further confirmed the significant role of BET surface area, morphology structures and pore size distributions in increasing the adsorption rate. And the corresponding descriptions of these values have been added in this manuscript (Please see Line 391-403, Page 11). Thanks again.

  1. In the Figure 2 the images d, e and f are not clear enough. Please present a better resolution of these images and for the images in Figure 3 and 5, also.

Reply: Thanks for your comment about this issue. These figures including Figure 2d, 2e, 2f and Figure 5 illustrated the high resolution transmission electron microscope images, which can be relatively clearly observed the lattice spacings of samples. Moreover, these high resolution images are the clearest pictures we can provide on account of the testing instrument and sample itself. Hence, we did not provide better resolution of these images. Thanks again.

  1. Please specify what represent the following parameters: the saturation magnetization (Ms), remanent magnetization (Mr), and coercivity (Hc), values that were shown in Table 1.

Reply: We appreciate your great comment about this issue. The saturation magnetization (Ms) represents the maximum magnetization that a magnetic material can achieve when magnetized in an external magnetic field. Remanent magnetization (Mr) donates that the magnetic material still maintains a certain magnetization in the direction of the original external magnetic field after removing the external magnetic field when the magnetic material is magnetized to saturation. The coercivity (Hc) represents that magnetic field intensity is required to reduce its magnetization to zero, after the magnetic material has been magnetized to magnetic saturation. And the corresponding descriptions of these parameters have been added in this manuscript. (Please see Line 346-351, Page 10). Thanks again.

  1. In Figure 9 a, c and d, both points and lines appear. In the legend it is not mentioned what the lines represent.

Reply: We appreciate your great comment about this issue. Actually, Figure 9 displayed the scatter plots of the adsorption data and the lines are the fitting curves based on the four kinetic models. And the corresponding legend has been added in Figure 9 and marked with BLUE words. Thanks again.

  1. Figure 10 shows three types of data, experimental (dotted), calculated according to the Langmuir model (continuous lines) and calculated according to the Freundlich model (broken lines). Please specify this in the figure (legend) and at the name of the figure. In the legend, you must specify that those are the experimental data, and in the name of the figure, insert a few words about the experimental points.

Reply: Thanks for your comment about this issue. We have specified the figure legend and detailed illustrations experimental points have also been inserted in this manuscript (Please see Line 497-500, Page 15, marked with BLUE words). Thanks again.

Reviewer 2 Report

Title: A generalized method for the synthesis of carbon-encapsulated Fe3O4 composites and its application in water treatment

Authors: Shengxia Duan , Jian Liu, Yanling Pang, Feng Lin, Xiangyan Meng, Ke Tang, Jiaxing Li

General Comments:

o  In this paper, a new method for preparation of C@Fe3O4 composites has been developed. The as-prepared materials was characterized by different techniques such as the field-emission scanning electron microscopy (FESEM) and transmission electron microscopy (TEM), X-ray diffraction (XRD), inductively coupled plasma-atomic emission spectroscopy (ICP-AES), SQUID, BET, and tested for removal of MB from industrial effluents.

o   The structure of the article fulfills the structure of a research article.

o   4 keywords are included by the author.

  • The Introduction section provide sufficient background information for readers in the immediate field to understand the problem that this study addresses.

o   The author present in the Materials and Methods section the reagents, the experimental method and the equipment used.

o   In the Results and Discussion section, the author present and interpret the results of the performed experiments.

o   The paper ends with the Conclusions part. In this section the author mentions the conclusions of the research study.

I suggest to Reconsider after Major Revisions for the following reasons:

  1. page 5, line 177: it is really the hemetine structure or the hematite structure?
  2. Please give the space group for all the compounds, not only for α-FeOOH.
  3. In Figure 7, two pictures are presented. What represents the second picture?
  4. Page 7, Table 1: please check carefully the Hc values the values on the graph sims to be similar;
  5. Despite the fact that the surface area of the samples is different, there is no differences into the adsorption capacities. Can you give an explanation for this behavior?
  6. The authors carefully analyzed the magnetic properties of the as synthetized samples, but it does not explain how these properties influence the adsorption capacity. What is the connection between the studied property (removal of MB) and the magnetic properties of the particles?  
  7. At page 10, lines 436-437, the authors assert that: “The C@Fe3O4 nanorod owned the largest adsorption capacity while C@Fe3O4 nanodisk owned the lowest adsorption capacity”, but at lines 447, we can find the next affirmation: “this factor makes no difference to their adsorption capacity”. Which conclusion is correct?

Author Response

In this paper, a new method for preparation of C@Fe3O4 composites has been developed. The as-prepared materials was characterized by different techniques such as the field-emission scanning electron microscopy (FESEM) and transmission electron microscopy (TEM), X-ray diffraction (XRD), inductively coupled plasma-atomic emission spectroscopy (ICP-AES), SQUID, BET, and tested for removal of MB from industrial effluents.

The structure of the article fulfills the structure of a research article. Keywords are included by the author. The Introduction section provides sufficient background information for readers in the immediate field to understand the problem that this study addresses. The author present in the Materials and Methods section the reagents, the experimental method and the equipment used. In the Results and Discussion section, the author present and interpret the results of the performed experiments. The paper ends with the Conclusions part. In this section the author mentions the conclusions of the research study. I suggest to reconsider after Major Revisions for the following reasons:

  1. In page 5, line 177: it is really the hemetine structure or the hematite structure?

Reply: Thanks for your comment about this issue. The main composition in both hemetine and hematite are α-Fe2O3. These two are just different expressions. Hence, we have revised the hemetine into hematite to express more professional in this manuscript. Thanks again.

  1. Please give the space group for all the compounds, not only for α-FeOOH.

Reply: We appreciate your valuable comment about this issue. And the space group for α-Fe2O3 and Fe3O4 has been added into this manuscript (Please see Line 198-199, Page 5 and Line 233-234, Page 6, marked with BLUE words). Thanks again.

  1. In Figure 7, two pictures are presented. What represents the second picture?

Reply: We are deeply sorry for the fault here, since these two pictures are the same. Hence, we have deleted one of the pictures. Great thanks for your reminder to us.

  1. Page 7, Table 1: please check carefully the Hc values. The values on the graph seem to be similar.

Reply: Great thanks for your comment about this issue. The Hc values of C@Fe3O4 composites nanorod, nanodisk and nanoparticle were 238.86 Oe, 285.59 Oe and 195.58 Oe. And the differences in Hc values were not clearly shown in Figure 7. Hence, we have enlarged the insert image to see the differences more clearly. Thanks again.

  1. Despite the fact that the surface area of the samples is different, there is no difference into the adsorption capacities. Can you give an explanation for this behavior?

Reply: We appreciate your valuable comment about this issue. The BET surface areas of three C@Fe3O4 composites nanorod, nanodisk and nanoparticle were 31.407 m2·g-1, 26.682 m2·g-1 and 27.003 m2·g-1, respectively. And the maximum adsorption capacities (Qmax) based on Langmuir model were 34.54 mg·g-1, 32.94 mg·g-1and 33.64 mg·g-1. The BET surface areas of three C@Fe3O4 composites have some small variations, which resulted in the minute differences in the adsorption capacities. Moreover, the adsorption performance is the result of many factors including morphology structures, pore size distributions and BET surface area. Hence, BET surface area does not take determinative effect in the adsorption performance. Thanks again.

  1. The authors carefully analyzed the magnetic properties of the as synthetized samples, but it does not explain how these properties influence the adsorption capacity. What is the connection between the studied property (removal of MB) and the magnetic properties of the particles?

Reply: Great thanks for your comment about this issue. The magnetic properties are the macroscopic expression of the sample microstructure, including particle size, morphology and the characteristics of micro-pores. The magnetic properties of C@Fe3O4 composites were discussed carefully mainly due to their different morphological structures. Moreover, there was no obvious correlation between the magnetic properties and adsorption performance of C@Fe3O4 composites. The magnetic properties of C@Fe3O4 composites might have an impact on MB removal, since magnetism would affect the agglomeration of nanoparticles. However, adsorption performance is the result of many factors including morphology structures pore size distributions and BET surface area. Hence, magnetic properties had little effect on MB removal by C@Fe3O4 composites. Furthermore, previous studies were rarely reported the effects of magnetic properties on contaminants removal mainly utilizing the magnetic separation. Additionally, investigations on the effects of magnetic properties on contaminants removal will be part of the future extension of our work based on the reviewer’s suggestion. Thanks again.

  1. At page 10, lines 436-437, the authors assert that: “The C@Fe3O4 nanorod owned the largest adsorption capacity while C@Fe3O4 nanodisk owned the lowest adsorption capacity”, but at lines 447, we can find the next affirmation: “this factor makes no difference to their adsorption capacity”. Which conclusion is correct?

Reply: Thanks for your comment about this issue. Actually, the two descriptions are not in conflict. The C@Fe3O4 nanorod owned the largest adsorption capacity while C@Fe3O4 nanodisk owned the lowest adsorption capacity. The difference in adsorption capacity is the result of morphology structures and BET surface area, but has little connection with pore size, since the three C@Fe3O4 composites nanorod, nanodisk and nanoparticle have similar pore size. Moreover, to make the descriptions more clearly, we have revised this sentence (Please see Line 493-496, Page 14, marked with BLUE words). Thanks again.

Reviewer 3 Report

The manuscript is interesting, and the data fully support the conclusion. Some suggestions are given here below and in the attached annotated pdf file.

Abstract “removal of MB from industrial effluents”: can other applications be envisaged beyond MB removal?

Line 89-101: some results are reported but in this part introduction the target of the work must be stated, not the results: please rewrite.

Line 132-134 “The C@Fe3O4 nanodisk and C@Fe3O4 nanoparticle 132 were also prepared using α-Fe2O3 nanodisk and commercially available α-Fe2O3 as the 133 precursors through the similar procedure as mentioned above.”: the role of the commercial samples is not clear and must be stated in the introduction

Line “obtained through the least square fitting method,” details of the LS fitting method must be given. A plot resulting from the LS must be given in ESI file

Line 257: “The existence of carbon 257 was further corroborated by the energy dispersive X-ray spectroscopy (EDS) analysis 258 (see ESI Figure S2).” Is C due to carbon only or typical contaminants? The iron oxide precursor EDS: is it showing no or less carbon?

Line 277 “he dozens of nanoparticles.” Why dozen? Is there any proof of this estimation?

Line 345 and 362 and 379 and 379 and 413 414 every equation in the manuscript: numerate the equations and refer to eq. number within the text

Line 376 “the practical rate” in which sense “practical”? Explain better

Figure 9 or Table 2: please provide Rsquare or other fitting parameter to evaluate the interpolation

An extensive revision of the English is needed and a non exhaustive list of suggestion is give in the annotated pdf file

After these modifications, the manuscript can be accepted for publication

Author Response

Referee 3:

The manuscript is interesting, and the data fully support the conclusion. Some suggestions are given here below and in the attached annotated pdf file.

  1. Abstract “removal of MB from industrial effluents”: can other applications be envisaged beyond MB removal?

Reply: We appreciate your valuable comment about this issue. This description illustrates the potential applications of the synthesized C@Fe3O4 composites. And investigations on other effluents, such as congo red, rhodamine B and heavy metal ions, are still in progress. Thanks again.

  1. Line 89-101: some results are reported but in this part introduction the target of the work must be stated, not the results: please rewrite.

Reply: Great thanks for your comment about this issue. Based on your comment, we have revised this part. Therefore, to protect Fe3O4 from being oxidized and apply suitable and “green” carbon sources, a simple and environmentally friendly method was developed for the preparation of highly stable dispersions of C@Fe3O4 composites with controlled morphologies, including nanorods, nanodisks, and nanoparticles, using sodium alginate as the carbon source and the easily obtained α-Fe2O3 as the precursors (Please see Line 96-99, Page 2 and Line 100-101, Page 3, marked with BLUE words). Thanks again.

  1. Line 132-134 “The C@Fe3O4 nanodisk and C@Fe3O4 nanoparticle were also prepared using α-Fe2O3 nanodisk and commercially available α-Fe2O3 as the precursors through the similar procedure as mentioned above.” The role of the commercial samples is not clear and must be stated in the introduction.

Reply: We appreciate your valuable comment about this issue. In particular, the commercially available α-Fe2O3 materials were also applied to serve as the precursor mainly based on the following considerations. Firstly, it is used to verify whether this method can be generally applied to prepare similar C@Fe3O4 composites with different kinds of precursors. Secondly, it is used to compare the adsorption performance of C@Fe3O4 composites for MB removal by using different kinds of precursors and further to confirm the morphology effect on the adsorption performance. Based on your comment, we have added this into our manuscript (Please see Line 108-114, Page 3, marked with BLUE words). Thanks again.

  1. Line “obtained through the least square fitting method,” details of the LS fitting method must be given. A plot resulting from the LS must be given in ESI file.

Reply: We appreciate your valuable comment about this issue. The cell parameters of materials synthesized in this study were obtained from the XRD analysis software, which directly illustrated the cell parameters by the least square fitting method based on the original XRD data. Hence, we mentioned that the cell parameters of α-FeOOH were obtained through the least square fitting method. And the least square fitting method for calculating the cell parameters by manual operation will be part of the future extension of our work. Thanks again.

  1. Line 257: “The existence of carbon was further corroborated by the energy dispersive X-ray spectroscopy (EDS) analysis (see ESI Figure S2).” Is C due to carbon only or typical contaminants? The iron oxide precursor EDS: is it showing no or less carbon?

Reply: We appreciate your valuable comment about this issue. Figure S2 illustrated the EDS examination of the C@Fe3O4 composites, mainly including elements of Fe, C and O. Specifically, C element is only ascribed to a simple substance instead of typical contaminants. Moreover, the iron oxide precursor contained both α-Fe2O3 and alginate beads, among which the carbon existed as constituent element of the mixture other than a simple substance. Carbon exists in two different forms in both C@Fe3O4 composites and iron oxide precursor. Furthermore, this study mainly discussed the structural compositions of C@Fe3O4 composites. Hence, the EDS of iron oxide precursor was not provided in this paper. Thanks again.

  1. Line 277 “the dozens of nanoparticles.” Why dozen? Is there any proof of this estimation?

Reply: Thanks for your comment about this issue. In general, the formation of mesoporous pores mainly includes the pore nature of the material itself and the aggregates among a lot of small nanoparticles. In this study, the pores in both C@Fe3O4 nanorod and C@Fe3O4 nanodisk were originated from the pore nature of the materials themselves, while the pores in C@Fe3O4 nanoparticles were originated from aggregates among a lot of small nanoparticles obtained from the observation of TEM images of the three C@Fe3O4 composites. We have revised this sentence in this manuscript to make the description more appropriate (Please see Line 298-302, Page 8, marked with BLUE words). Thanks again.

  1. Line 345 and 362 and 379 and 379 and 413 414 every equation in the manuscript: numerate the equations and refer to eq. number within the text.

Reply: We appreciate your valuable comment about this issue. According to your comment, all the equations were renumbered in this manuscript. Thanks again.

  1. Line 376 “the practical rate” in which sense “practical”? Explain better.

Reply: Great thanks for your comment about issue. Generally speaking, there exist two kinds of rate determining steps, i.e. liquid film diffusion model and intra-particle diffusion model. Supposing the three curves of the Boyd model plot crossed the origin, intra-particle diffusion was the rate-limiting step, otherwise liquid film diffusion was the rate-limiting step. However, as illustrated in Figure 9d in our study, none of the three curves of the Boyd model plot has crossed the origin, suggesting that liquid film diffusion was the rate-limiting step for MB immobilization onto the three C@Fe3O4 composites. And the fitting results achieved a good agreement with that of intra-particle diffusion model. Furthermore, “practical” herein was not appropriate because of its Chinglish expression. Hence, we deleted this word here. Thanks again.

  1. Figure 9 or Table 2: please provide Rsquare or other fitting parameter to evaluate the interpolation.

Reply: Thanks for your comment about this issue. Since the intra-particle diffusion and Boyd kinetic models have many parameters, we have added another table named Table S2 in our Supplementary Materials. Thanks again.

Table S2 Parameters of intra-particle diffusion and Boyd kinetic models for the adsorption of MB onto C@Fe3O4 composites.

Samples

models

intra-particle diffusion (g·mg·min-1/2)

Boyd model

C@Fe3O4

nanorods

Parameters

k1

4.742

k2

3.136

k3

0.3673

R2

0.9290

R2

0.9476

R2

0.9632

R2

0.7527

C@Fe3O4

nanodisks

Parameters

k1

2.394

k2

0.9490

k3

0.6778

R2

0.9572

R2

0.9649

R2

0.9995

R2

0.9613

C@Fe3O4

nanoparticles

Parameters

k1

7.771

k2

0.7173

k3

0.3460

R2

0.8464

R2

0.9888

R2

0.9733

R2

0.9938

  1. An extensive revision of the English is needed and a non exhaustive list of suggestion is given in the annotated pdf file.

Reply: We appreciate your valuable comment about this issue. We have consulted an English native speaker and the whole manuscript has been revised carefully, hoping the revised version will be up to the standards of Molecules. Thanks again.

Round 2

Reviewer 1 Report

In my opinion, the work entitled:

"A generalized method for the synthesis of carbon-encapsulated Fe3O4 composites and its application in water treatment" may be published in the journal Molecules.

The authors have made the recommended changes and corrections.

Reviewer 2 Report

The authors answered all the questions and modified the paper in accordance with the reviewer requirements.